# Where to start with climate-smart forest management? Climatic risk for forest-based mitigation

Natalie Piazza[1], Luca Malanchini[1], Edoardo Nevola[2], Giorgio Vacchiano[1]

[1] Department of Agricultural and Environmental Sciences, University of Milan, Milan, 20133, Italy

[2] World Wildlife Fund for Nature, Rome, 00198, Italy

*Correspondence to*: Natalie Piazza (natalkabro@gmail.com)

**Abstract.** Natural disturbances like windthrows or forest fires alter the provision of forest ecosystem services such as timber production, protection from natural hazards, and carbon sequestration. After a disturbance, forests release large amounts of carbon and therefore change their status from carbon sinks to carbon source for some time. Climate-smart forest management

may decrease forest vulnerability to disturbances and thus reduce carbon emissions as a consequence of future disturbances. But how can we prioritize the stands most in need of climate-smart management? In this study we adopted a risk mapping framework (hazard × vulnerability) to assess the risk to climate-related forest ecosystem services (carbon stock and sink) in forests prone to windthrow (in the Julian Alps, Italy) and forest fires (in the Apennines, Italy). We calculated hazard by using forest fire and windthrow simulation tools, and examined the most important drivers of the respective hazards. We then

assessed vulnerability by calculating current carbon stocks and sinks in each forest stands. We combined these values together with the calculated hazard to estimate "carbon risk" and prioritized high-risk stands for climate-smart management. Our findings demonstrate that combining disturbance simulation tools and forest carbon measurements may aid in risk-related decision making in forests, and in planning decisions for climate-smart forestry. This approach may be replicated in other mountain forests to enhance our understanding of their actual carbon vulnerability to forest disturbances.

## 1        Introduction

Natural disturbances are common in forest dynamics, disrupting forest cover, changing forest structure, and leading to forest succession (Dale et al., 2001). In Europe, the frequency, intensity, and severity of disturbances like wildfires, windthrows, and insect outbreaks are increasing due to changes in land use and climate (Collins et al., 2021; Grünig et al., 2023; Lozano et al., 2017; Patacca et al., 2023; Seidl et al., 2017; Senf et al., 2021; Senf and Seidl, 2021; Sommerfeld et al., 2018). In the future,

forest fires may impact regions previously not at risk due to dry, warm weather and earlier snowmelt (Westerling et al., 2006). Furthermore, storm intensity will increase as an indirect effect of warmer and moister atmosphere, higher updraft velocities, and slower storm movement, thereby increasing the duration of local storms (Kahraman et al., 2021). Such novel disturbance regimes may cause unprecedented changes at large scales, potentially altering the functioning of forest ecosystems and the services they provide (Thom and Seidl, 2016). For instance, bark beetle outbreaks or windthrow can significantly reduce timber

production and value, while forest fire may increase hydrological instability, disrupt habitats for forest biodiversity, and diminish recreation potential (Albrich et al., 2018). Moreover, regulating services like carbon sequestration could be slowed, halted or even reversed, temporarily shifting forests from sink to source (Albrich et al., 2022; Harris et al., 2021; Pugh et al., 2019; Yamanoi et al., 2015).

Forests contribute to natural climate solutions by acting as the largest land carbon sink (Griscom et al., 2017), storing carbon in their biomass, soil, and organic matter. Furthermore, they regulate the climate by influencing weather patterns, precipitation, and contributing to local and global cooling. To counter the increasing risks posed by forest disturbances to such contributions, climate-smart forestry (CSF) has been suggested as an approach that maximizes climate mitigation provided by the forest ecosystem and forest-wood products chain (Nabuurs et al., 2018). CSF aims to mitigate climate change by reducing greenhouse gas emissions and increasing carbon sequestration, creating more resilient forests through adaptive forest management, and enhancing productivity and the provision of other ecosystem services (ES; Nabuurs et al., 2018). Several case studies have demonstrated the positive effects of CSF on climate change mitigation, even when considering trade-offs between different ES (Górriz-Mifsud et al., 2022; Gregor et al., 2022; Peltola et al., 2022).

One aspect of climate-smart forestry (CSF) is fostering adaptation to increased disturbances like wildfires and windthrows by promoting structural, physiological, and ecological factors of forest resistance or resilience (Nabuurs et al., 2018), such as by increasing species diversity (Seidl et al., 2011). Other climate-smart strategies include selective thinning, changing forest rotation lengths, and promoting continuous-cover forestry (Verkerk et al., 2020). The impact of different disturbances depends not only on the disturbance agent but also on the pre-disturbance forest structure (Vacchiano et al., 2016). Factors such as forest density, tree species composition, topography, and soil properties influence susceptibility to windthrow (Quine et al., 2021). In the case of wildfires, forest structure along with the amount of fuel and moisture affects forest flammability and the probability of fire spread (Agee, 1996; Varner et al., 2015).

The efficacy of climate smart forestry in regulating services can be assessed by calculating the additional carbon sink or avoided carbon emissions resulting from enhanced management relative to current practices. In the context of growing demand for ecosystem services and increased disturbance risk, it is of great interest to identify the most effective forest management strategies to enhance forest resistance and resilience and avoid or reduce disturbance-related emissions. However, prioritizing climate-smart forestry across entire forest catchments or ownerships can be challenging due to the multifaceted nature of disturbance hazards and the varying levels of ecosystem services provided by each forest stand at risk. In this study, we model the climatic hazards of windthrow and forest fire hazard in two Italian forests, assess the vulnerability of forest carbon stock and sinks to these hazards, and calculate the climate risk for these critical ecosystem services at the scale of individual forest stands. This workflow enables managers to prioritize high-risk stands for climate-smart management.

## 2 Materials and methods

### 2.1 Study areas

We chose two study areas to proceed with our analysis, one for windthrow and one for forest fire hazard. Fusine study area is located in the Julian Alps, NE Italy (region of Friuli Venezia Giulia, Fig.1a). It is characterised by typical alpine climate, with high annual average precipitation of 1545 mm and mean annual temperature of 8° C (Tarvisio weather station, data from 1999-2024; ARPA FVG, 2024). The studied area stretches from 750 m a.s.l. to 2200 m a.s.l. and the soil types are Cambisols, Leptosols and Luvisols. The dominant tree species are Norway spruce (*Picea abies*), European beech (*Fagus sylvatica*) and silver fir (*Abies alba*), and typical forest compositions here are: spruce-beech-fir, spruce, spruce-beech (the most common) and some stands of European larch (*Larix decidua*) and dwarf pine (*Pinus mugo* subsp. *mugo*; Fig. 1b). Forests at Fusine offer provisioning ES valuable for its timber production, but also cultural service as the lakes of Fusine are an important touristic destination. Some of the forest fulfil protective function against natural hazards. Fusine area features high forests managed with shelterwood system in groups and the single-tree selection cutting in uneven aged forests. Thinning from below in young forests is executed to favour the dominants (De Crignis, 2020). The forest management plan of Fusine reports only occasional small, concentrated windthrows affecting a limited number of plants. However, in the total harvested volume, extraordinary cuts (including salvage logging after windthrow and bark beetle together, and to a lesser extent cutting for road construction) amounted to 30% of the total harvested volume in the years 1997-2019. Between 2016 and 2019, no regular cuts were made, but a significant volume of 11,125 m³ of wood was harvested as extraordinary (De Crignis, 2020).

Galeata study area is situated in the northern Apennines, Italy (region Emilia-Romagna, Fig.1a). It has temperate oceanic sub-Mediterranean climate with low precipitation of about 630 mm and an average temperature of 12° C per year (Meldola weather station, data from 1989-2019). The forest grows from the valley bottom at 300 m a.s.l. to the mountain tops of around 1000 m a.s.l. The prevalent soil types are Cambisols and Regosols. At Galeata, tree species composition varies between different types of forest stands (Fig. 1c). There are coppice stands with tree species like hop-hornbeam (*Ostrya carpinifolia*), pubescent oak (*Quercus pubescens*), manna ash (*Fraxinus ornus*) or Turkey oak (*Quercus cerris*), protection forests of conifer afforestation with spruce, black pine (*Pinus nigra*) and Scots pine (*Pinus sylvestris*), and coppice stands in transition or high forests of hop-hornbeam, pubescent oak, chestnut (*Castanea sativa*), European beech, Italian maple (*Acer opalus*), field maple (*Acer campestre*), manna ash, true service tree (*Sorbus domestica*), wild service tree (*Sorbus torminalis*), whitebeam (*Sorbus aria*), wild cherry (*Prunus avium*) and Douglas fir (*Pseudotsuga menziesii*). Forests at Galeata offer an important regulating ecosystem service, protecting against soil erosion and regulating water supply. Additionally, they contribute to the cultural function for recreation and tourism. Forest management especially aims to reduce the fire risk by improving the state of the current coppice stands and converting some of them to high forests. In high forest stands (especially in the conifer afforestations) usual thinning operations are implemented. The Galeata study site is located in an area with a "moderate" forest fire risk index. However, although there are no significant fires reported in the area dating back to recent decades, given the

scarce rainfall, summer droughts, and often excessive tree density of forest stands, the fire hazard should be strongly considered in this area.

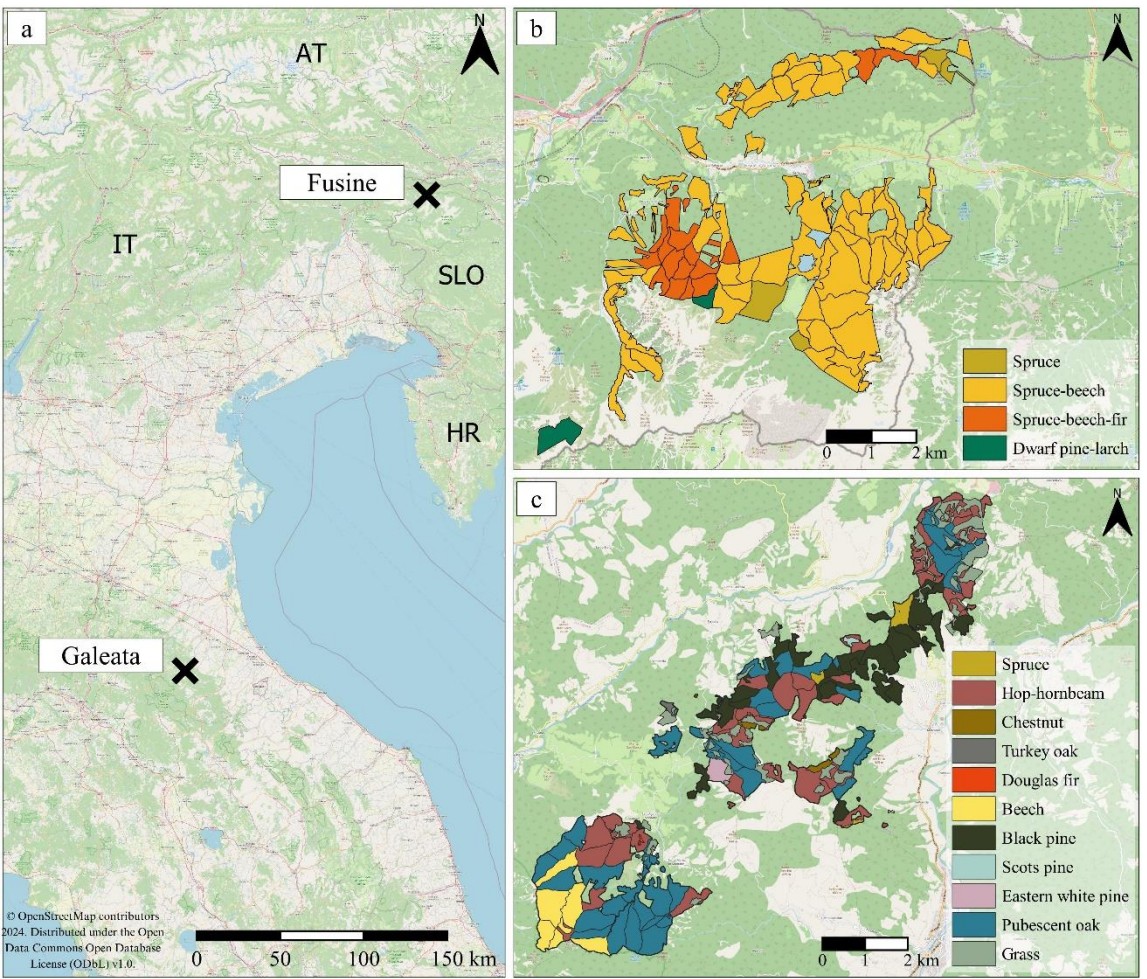

Figure 1: (a) Location of the study areas Fusine (Friuli Venezia Giulia) and Galeata (Emilia-Romagna). (b) Fusine represents typical montane to alpine forests stands in the Alps with dominating species spruce and beech. (c) Galeata has very heterogeneous forest stands from coppice dominated by broadleaves like hop-hornbeam and pubescent oak to conifer afforestation with pine and spruce. Base map: © OpenStreetMap contributors 2024. Distributed under the Open Data Commons Open Database License (ODbL) v1.0.

## 2.2 Field sampling

Forest management plans existed at both sites, with a dendrometric description and silvicultural indications for 98 forest stands in Fusine and 128 in Galeata. At Fusine, we used data from forest management plans to characterize the stands for ForestGALES simulations. Field data were used to validate the forest management plan data, which were collected using

LiDAR. At Galeata we used a combination of data from forest management plans (field-measured) together with our field data

(i.e., crown base height) to provide details per species and diameter class. The stands characteristics served as an input data for FlamMap simulations. To check that the data was up to date we measured forest structural parameters in 23 forest stands in Fusine and 45 in Galeata, i.e., 23% and 35% of all existing stands, respectively, making sure that all existing forest cover types were represented by at least five measured stands. For every stand, we randomly established a circular sampling plot with 10 m radius. We measured diameter at breast height (DBH) of all trees within each plot; we also recorded total height

and crown base height of three randomly chosen trees for each tree species and diameter class (<15 cm, 15-30 cm, >30 cm). If less than three individuals per species and diameter class could be found in a plot, we measured all trees available, as all data would then be pooled to build site- and species-specific height-diameter and crown ratio-diameter models. These trees were also sampled with an increment borer to estimate tree age and the width of the last ten annual rings from the increment core.

We carried out a sampling campaign to obtain the data on fuel loads at Galeata. We applied a standard scheme (Bovio and Ascoli, 2013) establishing two random sampling sites in each of four main forest types. Three transects of 10 m in form of an equilateral triangle were arranged at each site, with the following data being measured every 1 m of the transect: number of shrubs, shrub height, height of herbaceous layer, forest litter thickness, and humus thickness. Shrubs were measured only at 1 m intervals along the transect. If a shrub intersected the transect at more than one 1 m interval, it was counted as multiple

shrubs. All pieces of coarse deadwood (2.5 – 7.5 cm in diameter) were counted if they intersected with the transect line. We calculated the total amount of coarse deadwood in t/ha using the following equation and multiplying the volume by a specific density of 440 kg/m3 (1):

$$V = \frac{1.234 \times n \times d^2 \times a \times c}{L},$$ (1)

where $V$ is the volume expressed in m$^3$ha$^{-1}$, 1.234 is a constant, $n$ is the number of counted intersections, $d$ is the mean diameter

of the class (i.e., 5 cm), $a$ is a correction factor of 1.13 for diameters smaller than 7.5 cm (Brown, 1974), $c$ is the correction factor of the slope of the transect (Brown et al., 1982), derived from this equation (2):

$$c = \sqrt{1 + (\frac{\alpha}{100})^2},$$ (2)

where α is the slope inclination of the measured transect in % (Brown et al., 1982). The slope was extracted from an available digital terrain model (DTM).

Using the number of shrubs intersecting the transect we calculated shrub density per hectare. Using the shrub height recorded in the field we calculated the DBH by using the following equation (3):

$$DBH = a1 + e^{\frac{\ln(H) - \ln(b1) + b2 \times \ln(a2)}{b2}},$$ (3)

where H is the shrub height and a1, a2, b1 and b2 are species-specific coefficients (Evans et al., 2015). Individual shrub biomass was then obtained by applying species-specific allometric equations (Albert et al., 2014; Jenkins et al., 2004) that use

height and DBH. The individual biomass was then multiplied by shrubs density to obtain the per-hectare shrub loads. The

density of shrubs per ha was calculated as 10 000/d$^2$, where d is the mean distance between shrubs determined as the total distance of the three intersects (30 m) divided by the number of shrubs intersected.

In addition, three samples of duff, litter, live herbs, live shrubs, and fine deadwood (0.6 – 2.4 cm in diameter) were collected in squares of 40 × 40 cm located at the centre of each side of the triangle. These samples were oven-dried in a laboratory at a temperature of 105 °C for 24 hours and finally weighed. Dry fuel loads were then averaged per plot and scaled on a per hectare basis.

### 2.3 Disturbance hazard

#### 2.3.1 ForestGALES

We used ForestGALES to simulate windthrow at Fusine, considering topography, forest structure, and soil parameters. ForestGALES (Locatelli et al., 2017) calculates the probability of windthrow damage to individual trees and forest stands based on factors such as tree species, height, DBH, rooting depth (shallow/deep), topography, soil type, and current tree spacing. Topographic exposure (TOPEX) was calculated from a digital terrain model (DTM, 1 m resolution) and averaged across each forest stand. Tree and forest variables were derived from field measurements and gap-filled with up-to-date data from existing forest plans (validity 2021-2035; De Crignis, 2020). Soil type was derived from the soil map of Italy with a scale of 1:100000 (L'Abate et al., 2015).

ForestGALES calculates the critical wind speed at which trees could be damaged by uprooting or stem breakage. We developed a synthetic windthrow hazard indicator by averaging the critical wind speed at 1.3 m height and at crown height for breaking and for overturning. For two-species stands, simulations had to be performed for each tree species separately in order to obtain species-specific critical wind speeds. Critical wind speed for the whole stand was then determined as the lowest critical wind speed per stand (CWS for Norway spruce).

We used the Global Wind Atlas to obtain the Weibull parameters A and K for each of the forest stand at Fusine. We used the Weibull parameters to determine a probability distribution of 1-hr wind speeds and the probability of exceedance associated to each critical wind speed from ForestGALES.

To analyse the role of forest parameters in influencing the vulnerability of forest stands to windthrow, we performed a multiple linear regression of average critical wind speed as a function of tree density, number of trees, relative species composition by volume, DBH, tree height, and height-to-DBH ratio. We applied the gamma distribution since critical wind speed values are strictly positive. Larch and Scots pine stands were excluded due to their small sample size. As the proportion of beech was correlated with that of spruce, we included only beech data in the final model. We performed model selection using AIC-based stepwise backwards selection.

### 2.3.2 FlamMap

We assessed fire hazard using burn probability (i.e., the likelihood of future fire occurrence) considering topography, fuel loads and weather using the simulation tool FlamMap. The FlamMap (Finney, 2006; Stratton, 2006) fire mapping and analysis system calculates fire behaviour characteristics (e.g., rate of spread, flame length, fireline intensity) and burn probability for surface and crown fire for each pixel within the landscape, based on topography, moisture, weather, forest structure, and fuel conditions, provided each as a raster layer, and a vector layer with user-defined ignition locations. Slope, elevation, and aspect were extracted from a digital terrain model (DTM) and resampled at 10m resolution. The main weather variable required by the simulations is wind, for which we used a software feature from FlamMap called "Wind Ninja" that uses weather and geographical information to calculate a raster of wind direction and speed for the whole study area. The input data included air temperature (Zepner et al., 2021), cloud cover (personal communication from local expert), longitude, and time zone. Input data on forest structural parameters include tree height, crown base height, and tree cover density. Species-specific, third-degree polynomial regressions were fitted between DBH and height and between height and crown base height measured in the field. We calculated tree height and crown base height for all trees in each plot, and averaged them to obtain estimates for each measured forest stand. For unmeasured forest stands, we filled in the average tree height and crown base height calculated from all other plots in the study area belonging to the same forest type. All 10 m within each forest stand was then assigned the same values of forest structural variables. Tree cover density was estimated from the High-resolution Copernicus layer on Tree Cover Density 2018, with a resolution of 10 m (European Environment Agency, 2020).

Fuel data must be provided in the form of standard fire behaviour fuel model (Scott and Burgan, 2005); fuel loads measured in the field for five load components (1h – duff and litter, 10h – fine woody debris, 100h – coarse woody debris, live herbaceous, and live shrub fuels) were compared to fuel loads of standard fuel by Scott and Burgan (2005) to assign the closest-matching fuel model to each landscape pixel, at the resolution of $10 \times 10$ meters. For other forest types that were not included in the field sampling, we used a surface fuel dataset for Italy (Ascoli et al., 2020). The fuel loads were compared to the standard fuel models, and associated with the standard model that best reflects the characteristics measured. The degree of humidity was set to low as we intended to simulate a rather extreme fire weather scenario.

As a wildfire hazard indicator, we chose burn probability (BP), which has been widely used in assessing wildfire hazard in forest management plans (Benali et al., 2021). To calculate the burn probability, we instructed the software to generate 500 random ignitions points in the study area and we set the maximum simulation time in 10 hours.

We then modelled BP as a function of FlamMap input data, to find the most important drivers of fire hazard. As many input variables were collinear, we decided to use regression trees to recognize the most important variables. We included the following input parameters: forest type (coniferous/broadleaves), tree height, crown base height, canopy bulk density, crown cover, elevation, aspect (cosine-transformed) and slope.

## 2.4 Vulnerability and risk of carbon stocks and sinks

We used allometric methods in order to calculate the amount of carbon stocked at the two study areas. Using DBH and tree height collected from field measurements and forest management plans, we calculated aboveground tree biomass (AGB) using species-specific allometric equations for Italy (Tabacchi et al., 2011).

$$AGB = V_{AG} * c * \rho * A, \tag{4}$$

where $V_{AG}$ is the aboveground volume, c is the conversion factor, $\rho$ is wood basal density [t m$^{-3}$] and A is the area [ha].

Biomass was converted into carbon stock using a carbon density value of 0.47. We then estimated belowground, deadwood, litter and soil carbon using empirical equations from the Italian National Forest Inventory (Vitullo et al., 2007). These algorithms, which are currently used to provide estimates on carbon stocks for the national inventory report of greenhouse gas emissions and removals, estimate carbon stocks in compartments other than aboveground biomass using an allometric approach. Belowground biomass (BGB) is calculated by applying species-specific, multiplicative root-shoot coefficients (R) to aboveground biomass following this equation:

$$BGB = V_{AG} * \rho * R * A \tag{5}$$

Deadwood biomass (DWB) is estimated as the product of aboveground biomass and dead-live mass ratios factors ($c_D$) recommended by the IPCC Guidelines for National GHG Inventories (i.e., 0.20 for conifers and 0.14 for temperate deciduous species):

$$DWB = V_{AG} * c * \rho * c_D * A \tag{6}$$

Finally, litter and soil (0-30 cm) carbon are estimated from aboveground carbon using a linear regression, calibrated on more than 7000 nationally available forest inventory plots (details can be found in Federici et al., 2008).

In order to calculate carbon sink, we used the increment cores collected in the field. Cores were mounted and sanded in the lab following standard dendrochronological methods, then scanned for subsequent analyses. We measured the total width of the last ten rings of each tree core using CDendro and CooRecorder (Cybis Elektronik & Data AB). We converted diameter increment into DBH and tree height time series using previously fitted DBH-height equations, and then to aboveground biomass and carbon increment per decade by applying allometric equations.

In order to evaluate the risk, i.e., the product of hazard and vulnerability, we rescaled the values for windthrow/fire hazard and carbon stocks and sinks from 0 to 1. The windthrow hazard was expressed in terms of critical wind speed, where the lowest values led to higher probability of windthrow hazard and therefore had to be rescaled from 1 to 0. The vulnerability of carbon stock and sink to either a windthrow or wildfire hazard was calculated as the product of rescaled hazard value and rescaled vulnerabilities (carbon stock and sinks). The highest values indicate the highest vulnerability for carbon stock or sink to windthrow or wildfire damage, for a total of four risk maps.

For all statistical analyses we used R software version 4.2.1 (R Core Team, 2022) together with RStudio version 2023.09.0 (RStudio Team, 2020). The visualisation was performed using the R package *ggplot2* (Wickham, 2016).

## 3    Results

### 3.1    Windthrow simulations

The simulated critical wind speeds ranged between 11 to 30 m s$^{-1}$ with a mean critical wind speed of 17.3 m s$^{-1}$ across the entire study area of Fusine. Combined with the Weibull parameters, the calculated probability of exceedance (Fig. A1 in the Appendix) identified a few stands with higher vulnerability to windthrow (shown in yellow and red in Fig. 2). These stands typically had a higher share of Norway spruce (70-95%) and greater slenderness (the ratio of height to DBH).

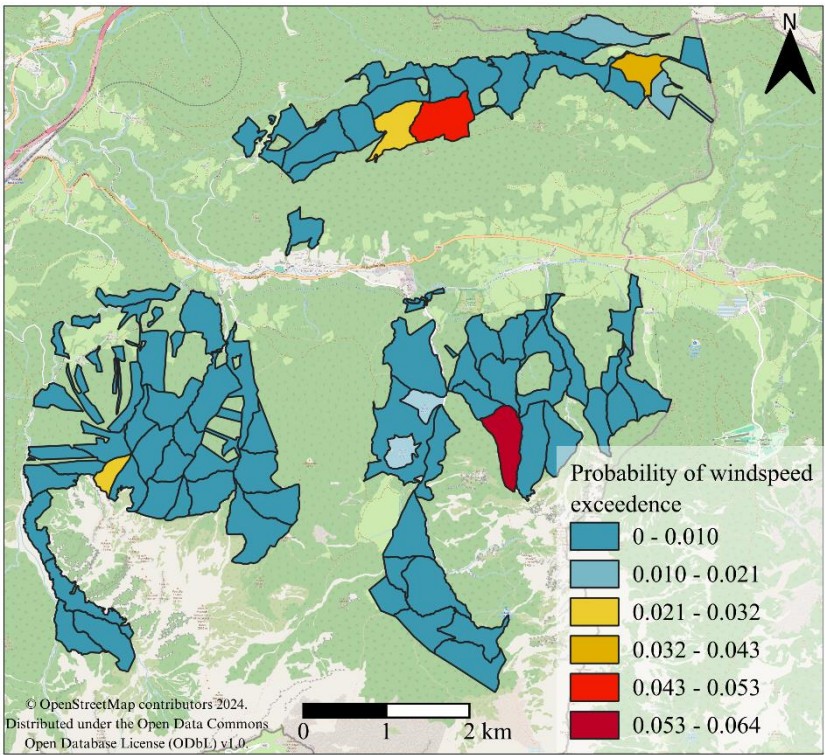

**Figure 2: Probability of wind speed exceedance calculated from the simulated critical wind speed (ForestGALES) and the Weibull parameters varied between 0 (blue) and 0.064 (dark red). The maximum values were reached in forests stands dominated by Norway spruce. Base map: © OpenStreetMap contributors 2024. Distributed under the Open Data Commons Open Database License (ODbL) v1.0.**

Density, share of beech (as percentage of volume), tree height, and slenderness were key variables influencing windthrow
exposure (Table 1). A higher proportion of beech relative to spruce were associated with higher critical wind speed (Fig. 3). Additionally, greater density and lower slenderness led to higher critical wind speed, indicating a less vulnerable and more stable forest stand.

**Table 1: Results of regression analysis with gamma distribution to predict the critical wind speed at Fusine.**

|  | Estimate | Std. Error | t value | Pr(>|t|) |
|---|---|---|---|---|
| Intercept | 24.159 | 3.367 | 7.175 | < 0.001 |
| Tree density | 0.024 | 0.002 | 13.465 | < 0.001 |
| Share of beech | 3.552 | 1.146 | 3.098 | < 0.01 |
| Slenderness | -20.362 | 2.945 | -6.915 | < 0.001 |

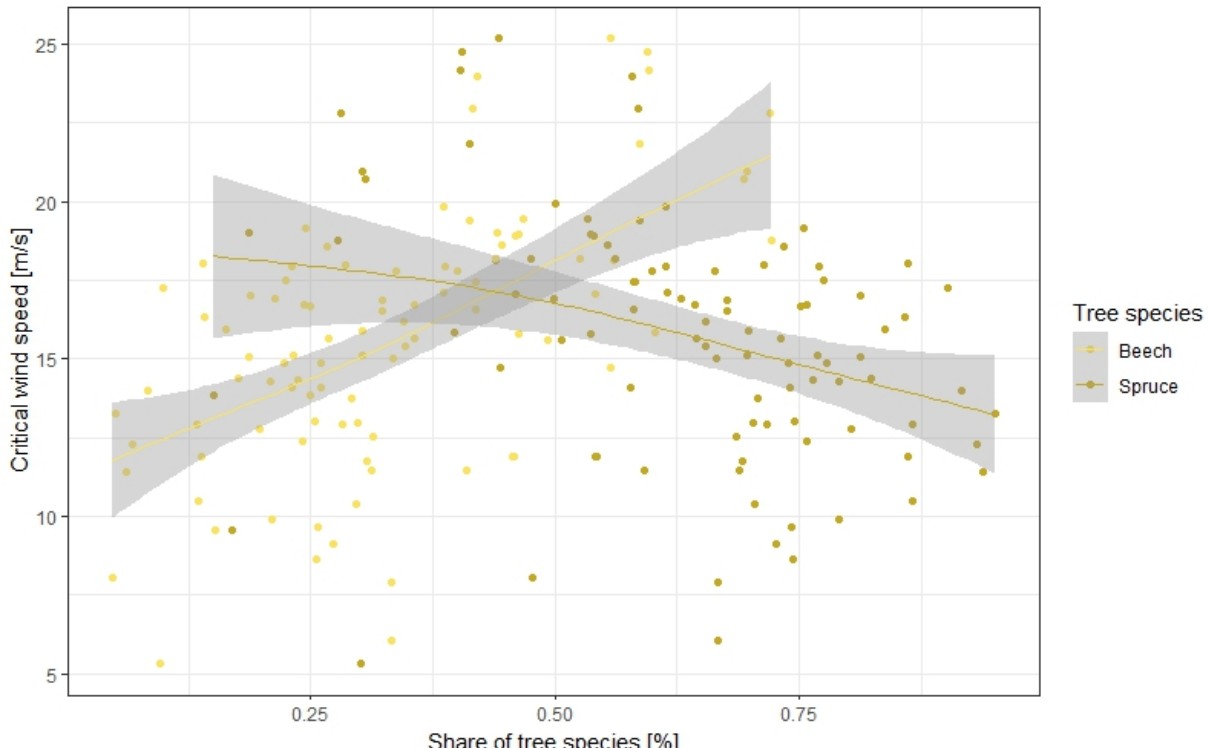


**Figure 3: Critical wind speed increases with higher stock of beech within a forest stand. Higher share of spruce increases the vulnerability to windthrow.**

### 3.2    Forest fire simulations

Fuel loads measured in the field varied among the main forest types (Table 2). The depth for pine forest type were similar as
in the hornbeam, but the fuel loads were lower with an exception of herb layer that was higher in pine forest type. Pubescent
oak had the highest fuel loads for 1-h, 10-h and shrub layers. The spruce forest type had the lowest values in the depth of the
overall flammable layer and fuel loads for duff, herb and shrub layers.

We selected three standard models to represent the forest types at Galeata and compared them to the calculated fuel loads and
the surface fuel dataset (Table 3 and Table A1 in the Appendix).

**Table 2: Fuel loads that were manually calculated from the sampling campaign for the main forest types.**

| Forest Type | Depth (cm) | Fuel load (t ha⁻¹) | | | | | |
| --- | --- | --- | --- | --- | --- | --- | --- |
| | | Duff | 1-h | 10-h | 100-h | Herb | Shrub |
| Pubescent oak | 9.54 | 11.95 | 8.48 | 2.60 | 3.94 | 0.59 | 1.60 |
| Hornbeam | 10.83 | 16.44 | 5.36 | 2.46 | 7.86 | 0.95 | 0.75 |
| Pine | 10.48 | 11.96 | 5.06 | 1.05 | 2.79 | 3.29 | 0.15 |
| Spruce | 6.49 | 6.06 | 7.42 | 2.48 | 8.65 | 0.19 | 0.06 |

**Table 3: Association between forest types in Galeata and the standard fuel models from** Scott and Burgan (2005)**.**

| Forest Type | Associated Standard Fuel Model |
| --- | --- |
| Pubescent oak | 164 |
| Hornbeam | 164 |
| Pine | 186 |
| Spruce | 186 |
| Chestnut | 165 |
| Turkey oak | 164 |
| Douglas fir | 186 |
| Beech | 164 |

The burn probability varied across different forest stands at Galeata and was highest in north of the area (Fig. 4). Forest stands that are more prone to fire were dominated by pubescent oak and hop-hornbeam.

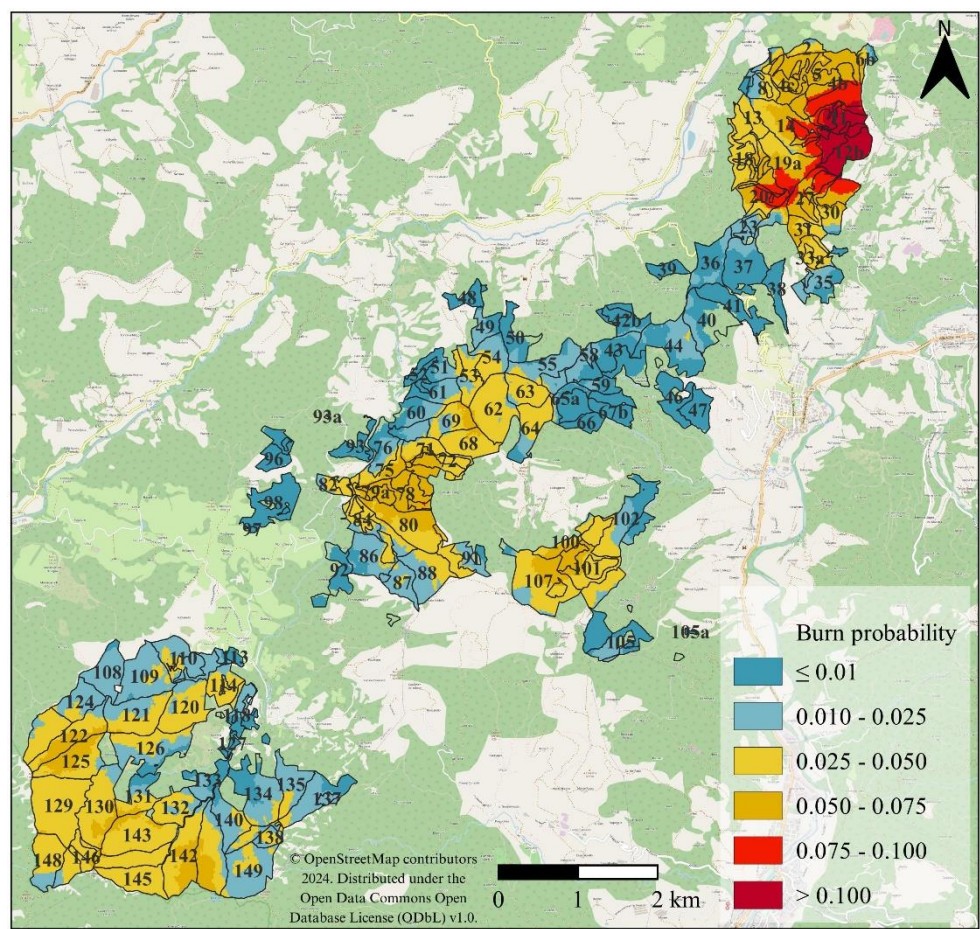

**Figure 4: Simulation results of burn probability from FlamMap. Base map: © OpenStreetMap contributors 2024. Distributed under the Open Data Commons Open Database License (ODbL) v1.0.**

The regression tree shows that the most important factor in determining burn probability is the forest cover type (coniferous vs. broadleaves; Fig. 5). In forest stands dominated by coniferous species (pine, spruce or Douglas-fir), the calculated fuel loads were the lowest, resulting in a lower fire hazard compared to broadleaved-dominated forests. Additionally, elevation and aspect significantly influenced fire susceptibility, with forests at lower elevations and on slopes with aspects from east through south to west being more susceptible. Higher crown cover further increases burn probability, as denser crowns spread potential crown fires faster and produce more litter on the forest floor.

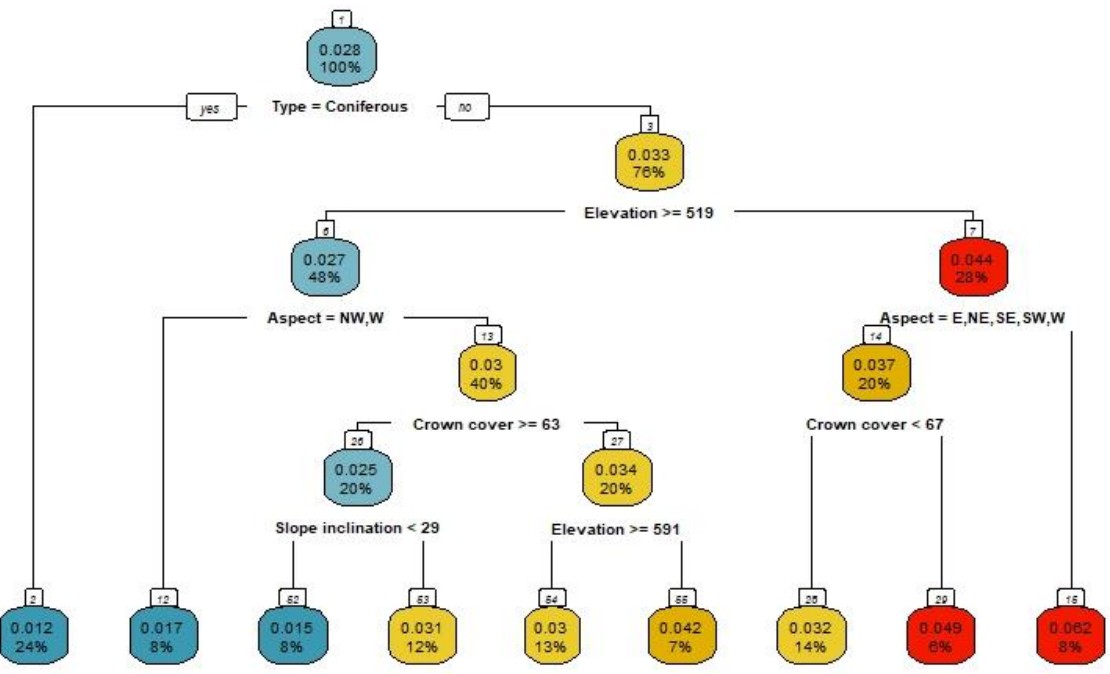

**Figure 5: Regression tree to predict the burn probability at Galeata. Nodes 2 and 3 were formed by splitting node 1 on the predictor variable *Type*. Node 2 consists of all rows with the value of *Type* = coniferous (24%) and node 3 consist of all rows with *Type* = broadleaves (76%). Next nodes are *Aspect*, *Crown cover*, *Slope inclination* and *Elevation*.**


### 3.3     Carbon stock and sink

The amount of carbon stock and sink varied by study area and forest stand (Fig. 6). The C stock at Fusine was greater compared to Galeata, but the C sink reached overall higher values at Galeata. The average carbon stock and $CO_2$ sink were 302 Mg ha$^{-1}$ and 9 Mg ha$^{-1}$/year at Fusine and 115 Mg ha$^{-1}$ and 15 Mg ha$^{-1}$/year at Galeata.

The final maps of carbon risk are reported in Figure 7. More vulnerable forest stands shown in darker colours and are a result of either high disturbance hazard (windthrow or wildfire) or high vulnerability of carbon stock/sink, or a combination of both.

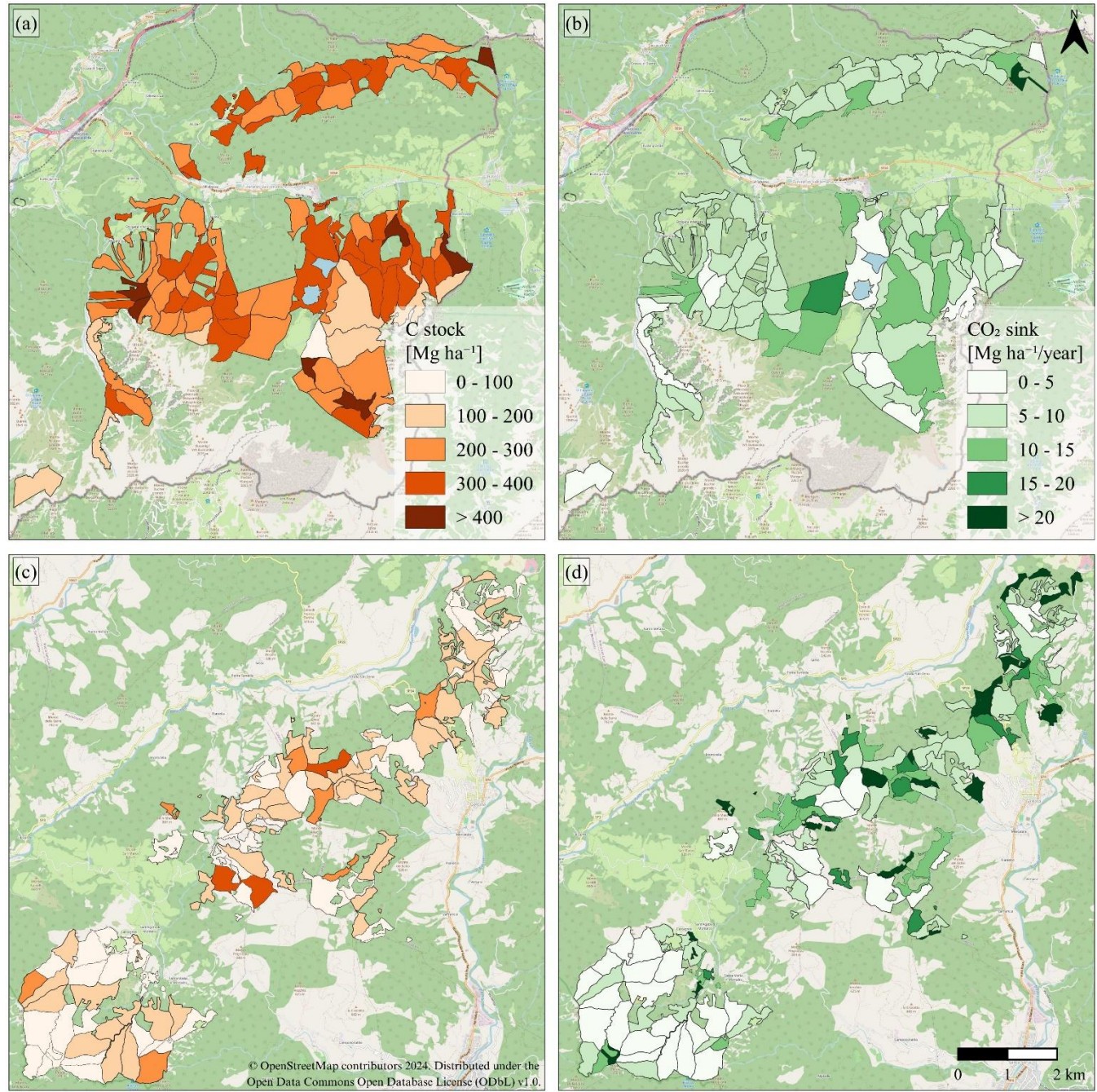

**Figure 6: The amount of C stock at Fusine (a) and Galeata (c) and CO₂ sink at Fusine (b) and Galeata (d). Base map: © OpenStreetMap contributors 2024. Distributed under the Open Data Commons Open Database License (ODbL) v1.0.**


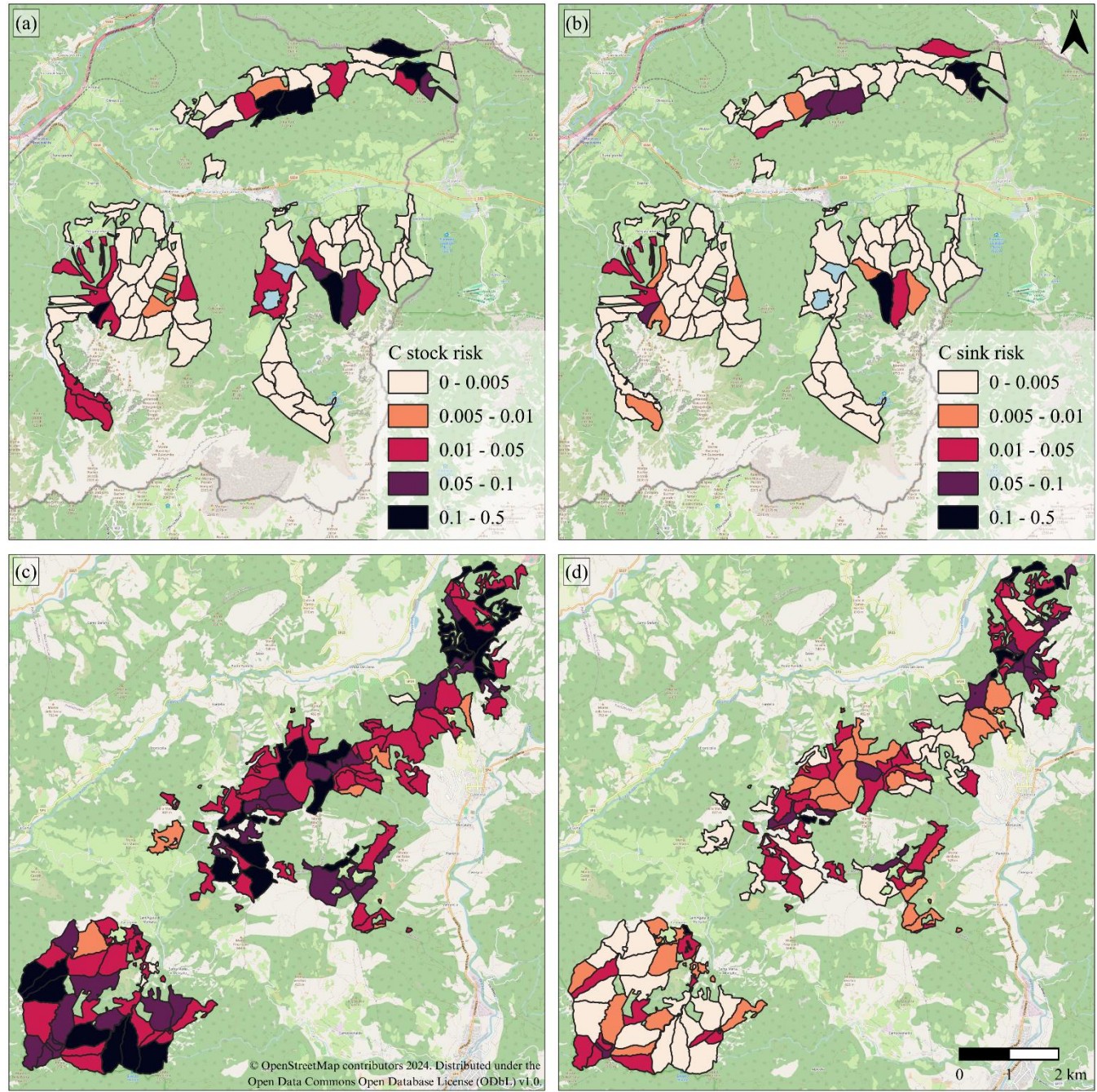

**Figure 7: The calculated risk to C stock (a) (c) and sink (b) (d) at the two study areas Fusine and Galeata. The darker the colour, the higher the risk. The C risk was calculated as the product of the amount of C stock or $CO_2$ sink and the exposure to either windthrow or wildfire. Base map: © OpenStreetMap contributors 2024. Distributed under the Open Data Commons Open Database License (ODbL) v1.0.**


# 4 Discussion

## 4.1 Disturbance-related risk for carbon stocks and sinks

Risk assessment and prevention for fire or windthrow hazard focuses on the human component of vulnerability. However, forest ecosystem services (ES), such as climate change mitigation, hydrogeologic protection or recreational value should also be considered as vulnerable assets. In this study, we have coupled state-of-the-art forest disturbance simulation tools with quantitative assessment of forest carbon stocks and sinks to prioritize risk prevention strategies in mountain forests. Climate-smart forestry may help to reduce forest disturbances leading to mitigation of carbon loss. The combination of the hazard analysis together with the vulnerability to carbon allowed us to identify forest stands with higher carbon risk. Therefore, proper silvicultural interventions were performed in these areas. Prevention silviculture aims to enhance those characteristics of forest stands that increase the resistance and resilience of forests to the hazards analysed, and, consequently, minimize the loss of ES. Different management strategies together with future climate will affect the temporal stability and the level of ES provisioning. However, in case of increased temporal stability of ES, the level of ES provisioning may be lowered, so an acceptable trade-off should be found for each local case (Albrich et al., 2018).

Data collection for simulating the provision of various ES may be based on different data sources from fieldwork or existing forest management plans as in our study, or from available digital data derived from remote sensing. These may include vegetation height models, digital terrain or surface models (Brožová et al., 2020, 2021). Such data vary in resolution based on the platform and sensor used, and provide information on crown coverage, canopy gaps, surface roughness, tree height, and other forest structural parameters. Such parameters may be used to simulate disturbance severity and probability of occurrence, and expected forest growth using forest dynamics models. Forest models are useful for assessing the impacts of climate change or different management scenarios on future levels of ES (Albrich et al., 2018), and to quantify carbon amounts "saved" by preventive silviculture as discussed in this paper.

Our methodology calculates carbon risk as a combination of disturbance hazard analysis together with carbon stock and sink exposure. Forest disturbance may be simulated using hybrid-mechanistic models (e.g., ForestGALES and FlamMap models used here) or statistical models, such as machine learning (Hart et al., 2019; Pawlik and Harrison, 2022). Statistical models require information about the observed damage and are therefore not suitable if this kind of data is not available. We showed that hazard modelling can be a valuable support to forest planning and management in order to maintain or improve the provision of ecosystem services. However, simulation tools for disturbance hazard assessment may not be easily used by forest practitioners. Therefore, we analysed the most predicting parameters for windthrow and wildfire hazards. Forest parameters for both disturbance hazards may be easily obtained from databases like forest management plans or available digital elevation models. For windthrow hazard, structural parameters like density, and slenderness, together with species composition, were found to be the most important determining the vulnerability. Denser forests with higher slenderness and higher share of spruce led to lower critical velocities, i.e., a higher probability of windthrow. With increasing tree height, there is a corresponding increase in the force exerted on the tree at a given wind speed. This increased susceptibility to force makes taller, slender trees

more prone to windthrow. Indeed, slenderness is one of the most frequently used indicators to estimate the stability of trees to strong winds (Hanewinkel et al., 2013). Analysis of satellite images supports our results, where natural disturbance like windthrow and bark beetle outbreaks occurred mainly in denser spruce forests (Stritih et al., 2021). In a study from Germany, tree height and tree species were found to be the most important predictors for windthrow damage. Similar to our findings, coniferous species like Norway spruce were the most vulnerable tree species (Albrecht et al., 2012). In our case, wildfires were

best predicted by forest type (coniferous/broadleaves), elevation, aspect and crown cover. Different forest types influence the probability of fire through various fuel content and also different amounts of moisture. Southern and western aspects have warmer and drier conditions due to greater solar radiation during the day and are thus more fire-prone compared to other aspects (Pandey and Ghosh, 2018). Likewise, meteorological factors influence the probability of forest fire significantly, especially temperature and relative humidity (Zhang et al., 2023b). Using these forest parameters together with past event

documentation may help forest practitioners to better identify the spatial extent of hazard in the mountain forests. The value of the regression models lies in their potential applicability to other areas without the need to run detailed risk models, which require using and understanding specific software. This could provide a practical tool for broader forest management strategies, enabling effective risk assessment and management in diverse forested regions. While the study focuses on specific stands, this approach could be applied to map risks at larger scales. This allows for a broader application in regional and national forest

management planning, offering a comprehensive tool for assessing and managing forest vulnerability.

Despite the overall low sequestration rates compared to the anthropogenic carbon production, mountain forests are more efficient in carbon sequestration in comparison to lowlands, (Schirpke et al., 2019). In case of disturbance, the in-situ C sink may shrink (Lindroth et al., 2009) or even turn to C source. Our methodology provides a good basis for calculation of forest carbon scenarios in case of improved forest state and thus a decrease in forest damage in the future. Furthermore, it is possible

to use our data to calculate the amount of carbon credits that may be generated in the respective areas. However, our methodology only considers aboveground, belowground, deadwood, litter and soil carbon and does not account for the respiration of the forest and soil, including root and microbial respiration. In a spruce forest, soil respiration is about 50% the carbon gain, equally divided between root and microbial respiration (Schulze, 2006).

We present both carbon stock as well as carbon sink as potential assets at risk. However, it is necessary to acknowledge the

importance of carbon stocks representing long-term carbon reserves that would require significant time to rebuild in case of their loss. Thus, maintenance of carbon stocks should be prioritized, since their reduction (through e.g., a disturbance) may lead to long-term negative impacts on carbon balance (e.g., Law et al., 2004). Harvesting high-risk forest stands with higher carbon stocks and storing carbon in harvested wood products could mitigate the actual risk while ensuring the carbon storage. A high carbon sink reflects a healthy, vigorously growing forest, which is important for continuous carbon sequestration.

Harvesting these forests and replacing them with slower-growing species would diminish their growth potential and overall carbon sequestration capacity, thus hindering climate mitigation efforts. In the framework of our study, it would be more effective to focus on high-stock forest stands at risk. Additionally, considering shortening the harvest period for younger forests

with high carbon sinks at risk may also be beneficial. This approach balances the protection of long-term carbon reserves with the maintenance of continuous carbon sequestration.

Finally, carbon loss mitigation is an important aspect of disturbance avoidance, but maintaining other ecosystem services may be just as important in the future. Mountain forests provide a variety of ES, both globally and locally, like protection against natural hazards, provision of drinking water, food and forage (Schirpke et al., 2019). Timber production has been historically one of the most important forest functions – it provided people with building material and energy source. Wood as building material is coming back to the focus as it provide a sustainable product substitution and a possible improvement of forest

carbon sinks (Kauppi et al., 2018).

## 4.2    Priorities for Climate Smart Forestry

In forest stands that were most prone for windthrow or wildfire, management interventions were proposed. The objective in windthrow-prone forests is to improve the resistance characteristics of forest stands to increase the critical wind speed that causes breakage or overturning. For simplification purposes, we did not account for different forest edges in our simulations,

setting this parameter to a constant – "windfirm edge." Nevertheless, we acknowledge that unstable edges may cause damage, and possible treatments include feathering (i.e., edge thinning to decrease wind loads and preserving trees that are more likely to withstand windthrow damage). In our case, improving forest edges would include favouring more resistant tree species like beech. It is also recommended to avoid a new forest edge, as the trees growing within the stand are less adapted to higher levels of exposure. Harvesting should avoid creation of new edges and use already established edges, as e.g., a forest road.

Forest stands should be managed to grow more stable trees with larger diameters and better root system, which can be a result of a good selective thinning (Mason and Valinger, 2013). However, selective thinning in spruce stands must be done in earlier stand age – forestry experts suggest before 30-50 years of stand age depending on the soil fertility. Otherwise, for a period of time, it destabilizes the treated forest stand that may completely collapse (Albrecht et al., 2012). We proposed an increased species diversity at Fusine, where broadleaved species may be favoured in unstable spruce-dominated forests to aim for better

stand stability (Nabuurs et al., 2018). Tree height may be decreased by shortening the rotation length (Albrecht et al., 2012), which also leads to a reduced probability of windthrow damage (Potterf et al., 2023). Furthermore, thinning operations, increased structural and age diversity, and avoidance of unstable edges were recommended. Such management strategies lead to higher temporal stability of ecosystem services provisioning as e.g., the C stock or timber volume (Albrich et al., 2018).

In the case of fire-prone forests, the aim is to reduce the flammability of stands through reduction of fuel load and continuity,

with the purpose of modifying fire behaviour to reduce the intensity of an eventual fire and the consequent loss of stand. The possible forest interventions include: opening of discontinuities to have more air flow and thus greater heat loss; selective thinning with larger diameters and with higher crown insertion; reducing the density and size of individual tree groups; reducing the amount of shrub and deadwood to decrease the rate of spread and intensity of the flame, and to prevent fire spread from the ground to the crowns. Similarly, we proposed management in forest stands at Galeata with further management of

continuous forest cover by segregating these stands. Some of the common climate smart forestry practices in wildfire-prone

areas include thinning and removal of excess fuel, e.g., through prescribed fire. Such practices may improve the health of the forest and its ability to resist and increase the resilience to face extreme weather events (Nabuurs et al., 2018). Thinning decreases canopy cover and tree density as well as competition, thereby increasing the availability of resources such as ground light radiation, water and nutrients and consequently, in proportion to the intensity of the intervention, more vigorous growth of the remaining individuals (Pretzsch, 2005). Furthermore, thinning operations increase woody increment, especially with higher intensities (Bianchi et al., 2010). The difference in increment may reach up to 20-30% more compared to non-thinned forest stands (Kim et al., 2016). As a direct consequence of increased growth increment, thinning allows stands to absorb more carbon from the atmosphere (Collalti et al., 2018). Comparing the carbon stocks 30 years after thinning with different intensities, no change in biomass has been observed and thus, also no difference in carbon stocks (Erkan et al., 2023). Thinning operations in secondary forests increased the tree diversity, which was positively correlated with carbon storage (Zhang et al., 2023a). These findings may bring good arguments for managing forest for an increased stability and biodiversity, making them more resilient and resistant to forest disturbances with no cost in the future biomass yields.

Implementing the suggested decision support tools will nevertheless represent a challenge in Italy, because of the scarcity of planned forests and ownership fragmentation. Eighty-five percent of forests currently lack a valid management plan. This is particularly true for small forest owners, as making their own plans is often too expensive. In order to deal with this issue, the Italian forestry law implemented the following: "To protect and manage agro-silvo-pastoral resources and improve abandoned lands, regions should promote formation of forestry consortia or cooperatives, enhancing the management of small properties and collective lands. These initiatives aim to reconstitute economically viable production units and foster new entrepreneurial activities (Testo unico in materia di foreste e filiere forestali, 2018)".

## 5    Conclusions

Implementing climate-smart forestry practices is crucial for minimizing forest disturbances and mitigating carbon loss. By integrating hazard analysis with carbon vulnerability assessments, it becomes possible to identify forest stands at elevated risk of carbon loss. In these areas, targeted silvicultural interventions are recommended to effectively manage potential carbon risk. We propose a methodology that combines disturbance hazard simulations with forest carbon exposure assessments to support risk-related decision-making and strategic planning for climate-smart forestry. This approach may be replicated in other mountain forests to better understand their carbon vulnerability to forest disturbances.

**Code and data availability**

Code and data are available from the corresponding author on request.

**Author contribution**

NP performed the statistical analysis and wrote the manuscript. LM performed the fire hazard simulations and wrote the manuscript. EN supervised the fieldwork, performed the analysis and wrote the manuscript. GV conceptualised the research, defined the research structure, wrote and reviewed the manuscript.

**Competing interests**

The authors declare that they have no conflict of interest.

**Acknowledgements**

The authors wish to thank Simone Cavenaghi, Luca Cirea and Vittoria Romagnano for the extensive work in the field and Federico Romanato for the help with ForestGALES simulations We thank local foresters for their collaboration: Gianluca Ravaioli and Andrea Ghirelli from Unione di Comuni della Romagna Forlivese (Galeata), and Gianfranco Dreossi and Gianluca Barnabà from Servizio foreste e Corpo forestale della Regione autonoma Friuli Venezia Giulia (Fusine). Thanks to 430 Michael Lombardo for proofreading this manuscript. This study was funded from the LIFE Programme of the European Union (LIFE19 CCM/IT/001201) with the project "Promotion and Enhancement of Forest System through different Ecosystem Services aimed to increased Carbon Stock" (LIFE CO2PES&PEF).

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

**Appendix**

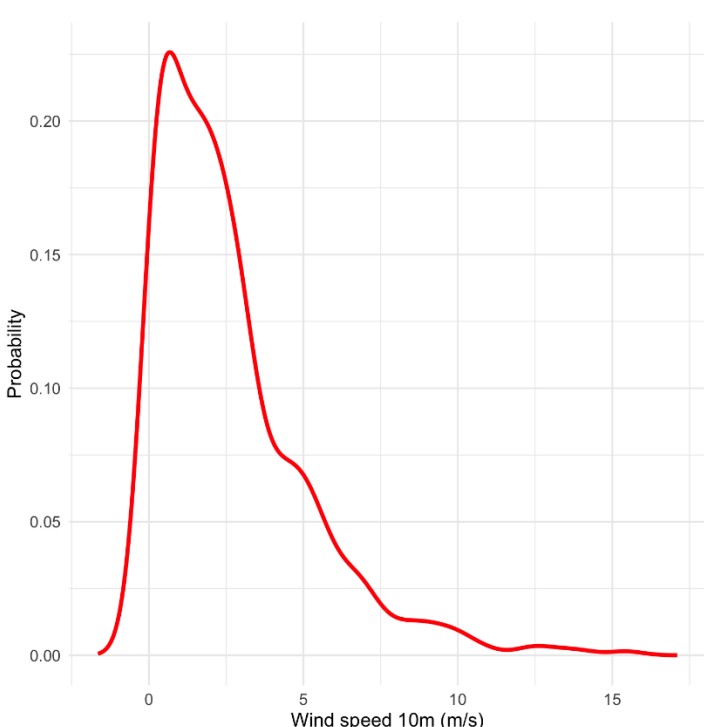


**Figure A1: The calculated probability of exceedance of wind speed at 10 m height.**

**Table A1: Fuel models chosen to represent forest stands at Galeata to simulate burn probability with FlamMap and the respective area with the respective standard fuel model.**

| Standard Fuel Model | Fuel Model Code | Fuel Model Name | Fuel load (t ha$^{-1}$) | | | | | |
|---|---|---|---|---|---|---|---|---|
| | | | 1-h | 10-h | 100-h | Herb | Shrub | Area (ha) |
| 164 | TU4 | Dwarf Conifer with Understory | 10.09 | 0 | 0 | 0 | 4.48 | 1484.45 |
| 165 | TU5 | Very High Load, Dry Climate Timber-Shrub | 8.97 | 8.97 | 6.73 | 0 | 6.73 | 16.93 |
| 186 | TL6 | Moderate Load Broadleaf Litter | 5.38 | 2.69 | 2.69 | 0 | 0 | 444.47 |
