# Peer review of "Where to start with climate-smart forest management? Climatic risk for forest-based mitigation"

_EGUsphere, 2024_

## Author Response (AR1)

**Dear reviewer 1, we appreciate the valuable feedback and constructive comments provided.**

**You can find our responses to your comments in bold text.**

General comments

I think this manuscript is a good example of how to leverage field-based information and disturbance simulation tools, to inform decision-making in forest management under risks connected to global change, and represents a promising way forward in climate-smart forestry, though its actual implementation might still find obstacles related to the socio-economic context.

Though I overall positively assess the study, I think the manuscript would greatly benefit from a thorough revision of the language by a proficient English speaker. What is pointed out in the "technical corrections" does not represent an exhaustive list, as I'm not a native speaker myself. Furthermore, a series of clarifications on the methodology is required before the publication.

**We carefully considered your suggestions and revised the manuscript to address the concerns, clarified the methodology and other issues including language revision.**

Specific comments

- In the section "Study area" you could more explicitly describe the typical disturbance regime of the two areas. Furthermore, add info on the climate station from which you got temperature and precipitation data, and specify to which interval of years these data refer to.

**We included the following two paragraphs describing the disturbance regime at the two areas:**

**The forest management plan of Fusine reports only occasional small, concentrated windthrows affecting a limited number of plants. However, in the total harvested volume, extraordinary cuts (including salvage logging after windthrow and bark beetle together, and to a lesser extent cutting for road construction) amount to 30% of the mass in the years 1997-2019. No ordinary cuts have been made in the years of 2016-2019, but there is a significant volume reported as extraordinary cuts.**

**The Galeata study site is located in an area with a "moderate" forest fire risk index. However, although there are no significant fires reported in the area dating back to recent decades, given the non-abundant rainfall, summer dryness, and often excessive tree density of forest stands, the fire hazard should be strongly considered in this area.**

**We added the information on weather stations and the respective years wo which these data refer to. For Fusine this is the weather station in Tarvisio referring to the data from 2003-2020. For Galeata study site we chose the weather station in Meldola that refers to the years 1989-2019.**

- In the field sampling section (lines 99-100), could you specify how the 3 trees per species and size were chosen? E.g. as the closest 3 to the plot centre? Furthermore, in cases when there were less than 3 trees of a species and size class, how did you proceed? You just sampled what you could find or got info from trees outside the plot?

**The text was modified as follows to clarify the issue:**

**We measured diameter at breast height (DBH) of all trees within each plot; we also recorded total height and crown base height (CBH) of three randomly chosen trees for each tree species and diameter class (<15 cm, 15-30 cm, >30 cm). If less than three individuals per species and diameter class could be found in a plot, we measured all trees available, as all data would then be pooled to build site- and species-specific height-diameter and crown ratio-diameter models.**

- Are management unit (section 2.2) and stand (2.3.1) equivalent? If so, why using different terminology?

**Forest management unit and forest stand are equivalent, we checked and corrected the use of these terms. We used only forest stand through the whole manuscript.**

- You mention "unmeasured forest units" at line 159, but earlier you wrote "For every unit, we randomly established a circular sampling plot with 10 m radius." (lines 97-8). Could you clarify how the unmeasured forest units came up?

**Some units remained unmeasured for logistic reasons. We edited the beginning of chapter 2.2 as follows for better clarity:**

**Forest management plans existed at both sites, with a dendrometric description and silvicultural indications for 98 forest stands in Fusine and 128 in Galeata. To check that the data was up to date and to complete dendrometric measures with estimates of carbon stock, sink, and vulnerability to hazards, we measured forest structural parameters in 23 forest stands in Fusine and 45 in Galeata, i.e., 23% and 35% of all existing stands, respectively, making sure that all existing forest cover types were represented by at least five measured stands.**

- Could you elaborate on why you applied different statistical models (multiple linear regression with stepwise backwards selection, and regression trees) to identify the main hazard drivers for wind and fire, respectively, and not the same type of model in both cases?

**We tested the multiple linear regression to identify the drivers for fire hazard. However, as many input variables were collinear, we decided to use regression trees instead, as described in the methods.**

- Could you further elaborate on how the estimation of "belowground, deadwood, litter and soil carbon using empirical equations from the Italian National Forest Inventory" was done (line 181-2)? Since they represent an important part of the carbon stock together with the aboveground tree biomass, I think it's important to be more detailed about the estimation methodology and its reliability.

We agree with the reviewer and added the following in order to clarify the issue.

**These algorithms, which are currently used to provide estimates on carbon stocks for the national inventory report of greenhouse gas emissions and removals, estimate carbon stocks in compartments other than aboveground biomass using an allometric approach. Belowground biomass (and hence carbon) is calculated by applying species-specific, multiplicative root-shoot coefficients to aboveground biomass. Deadwood biomass is estimated as the product of aboveground biomass and dead-live mass ratios factors recommended by the IPCC Guidelines for National GHG Inventories (i.e., 0.20 for conifers and 0.14 for temperate deciduous species). Finally, litter and soil (0-30 cm) carbon are estimated from aboveground carbon using a linear regression, calibrated on more than 7000 nationally available forest inventory plots (details can be found in Federici et al. 2008).**

*Federici S, Vitullo M, Tulipano S, De Lauretis R, Seufert G (2008). An approach to estimate carbon stocks change in forest carbon pools under the UNFCCC: the Italian case. iForest 1: 86-95. - doi: 10.3832/ifor0457-0010086*

- I think the role of microbial respiration, besides that of disturbances, in limiting forest's ability to be a carbon sink should not be overlooked, especially if this methodology would be used, as suggested, "to calculate the amount of carbon credits" (line 308). I think you should elaborate on this in the discussion.

*Furthermore, it is possible to use our data to calculate the amount of carbon credits that may be generated in the respective areas.* **However, our methodology considers aboveground, belowground, deadwood, litter and soil carbon and does not account for the respiration of the forest and soil, including root and microbial respiration. In a spruce forest, soil respiration is about 50% the carbon gain, equally divided between root and microbial respiration (Schulze, 2006).**

- Concerning forest management, I appreciated the identification of the main drivers of wind and fire hazards, to easily inform forest practitioners, and the part of the discussion including thorough management recommendations. I think what could be a good adding here are some considerations about the feasibility of implementing these decision support tools and related management strategies, given the scarcity of planning and ownership fragmentation in the Italian context.

**We agree with this problematic in Italy and added this to the discussion:**

**Implementing the suggested decision support tools will nevertheless represent a challenge in Italy, because of the scarcity of planned forests and ownership fragmentation. Eighty-five percent of**

**forests currently lack a valid management plan. This is particularly true for small forest owners, as making their own plans is often too expensive. In the last 30 years, policies were put forward to encourage joint or community management of small private forests, and combat fragmentation and abandonment of forest management. Such forest owner associations have different names, structure and objectives, but overall, they have the potential to improve the state of many forests in the country.**

Technical corrections

26 would use "furthermore" rather than "on the other hand", since the two sentences (about fire and wind respectively) are not in opposition, you're rather adding something to the previous sentence

**We agree and changed it accordingly.**

29 a better reference than Runkle 1985 here would be: Thom, D., & Seidl, R. (2016). Natural disturbance impacts on ecosystem services and biodiversity in temperate and boreal forests. Biological Reviews, 91(3), 760-781.

**We changed the suggested reference to Thom and Seidl (2016).**

34 Would remove "Indeed, "

**We removed it.**

43 Would use the word "strategy" instead of "goal"

**We have not found this in the manuscript. The sentence on line 43 goes:** *One aspect of CSF is to foster adaptation to increased disturbances as wildfires and windthrows by promoting structural,...*

50 it would be good to add a reference at the end of this sentence

**We added two references: Agee, 1996; Varner et al., 2015**

*Agee, J. K.: The Influence of Forest Structure on Fire Behavior, in: 17th Forest Vegetation Management Conference, 52–68, 1996.*

*Varner, J. M., Kane, J. M., Kreye, J. K., and Engber, E.: The Flammability of Forest and Woodland Litter: a Synthesis, Current Forestry Reports, 1, 91–99, https://doi.org/10.1007/s40725-015-0012-x, 2015.*

69, 83 Use the verb "offer" instead of "have" with (provisioning or regulating) ecosystem services

**Thank you for this suggestion, we agree.**

71 can be rewritten as "Fusine area features high forests managed with shelterwood system …"

**We changed this as suggested.**

111 no reference is reported for Brown et al., 1982

**We added this reference.**

*Brown J.K., Oberheu R., Johnston C. (1982), Handbook for inventory Surface Fuels and Biomass in the Interior West. U.S. Department of Agriculture, Forest Service, Intermountain Forest and Range Experiment Station. https://doi.org/10.2737/INT-GTR-129*

159 typo: tree height

**Corrected.**

161 typo: then

**Corrected.**

162 would use "resolution" instead of "grain"

**We changed it from grain to resolution.**

215 would use "varied among" instead of "differed in"

**Ok, it has been changed to "varied among".**

254 use "or" instead of "/" between C stock and $CO_2$ sink to avoid potential misinterpretation of that being a ratio

**Yes, we agree and we changed it to "or".**

261-2 rewrite as "… forest ecosystem services (ES), such as climate change mitigation, hydrogeologic protection and recreational value, should also be considered as vulnerable assets." I would argue that ES are, by definition, "of interest to society", so it's superfluous to repeat that.

**Ok, we have rewritten the sentence.**

295 add "found to be" as "tree height and tree species were found to be the most important predictors"

**Ok, we added it.**

299 use "greater solar radiation" instead of "more sun rays"

**Ok, it has been changed to "greater solar radiation".**

Dear reviewer 2,

**thank you for your thorough review and insightful comments on our manuscript. Your suggestions have been extremely helpful in improving our work, and we have incorporated your feedback into our revisions.**

**You can find our responses to your comments in bold text.**

The manuscript describes an attempt to operationalize the CSF concept for two areas in Italy, using state of the art disturbance models to identify high-risk areas.

Although I support the idea of the paper very much, I think there are several critical issues with the manuscript.

- The presented risk maps are relative, showing the areas where risks are higher than elsewhere. However, it would be very important to know the absolute risk to judge if action is really needed. If the absolute risk is lower than your tolerance level, no need to do anything. ForestGALES provides the critical windspeed, which is commonly used in combination with wind climate characteristics to judge what is the probability of exceedance of this critical windspeed.

**We agree to the reviewer's comment – the analysis on wildfire includes the calculation of the probability, but the windthrow hazard was missing the probability and we decided to add an analysis on the probability of the exceedance of the critical windspeed. We used the Global Wind Atlas to obtain the Weibull parameters A and K for each of the forest stand at Fusine. We used the Weibull parameters to determine a probability distribution of 1-hr windspeeds and the probability of exceedance associated to each critical windspeed from ForestGALES. We now show a new map of an absolute risk to windthrow at Fusine.**

- It is unclear how the combination of windspeed/fire risk and carbon stocks/sinks are made, what values result, and how they are grouped into classes. Only in the caption of figure 7 there is a remark that classes are "equally distributed by quantiles". This should be part of the methods. And again, this makes the classes relative, always showing 20% of your area at very high risk. Even if you would take action based on your analysis, there will always be 20% area at very high risk.

**Thank you for your feedback regarding the combination of wind/fire hazard and carbon stocks/sinks vulnerability. In the methods section, we detail the combination of windspeed/fire and carbon stocks/sinks. To evaluate risk, we rescaled the values for windthrow (using real exceedance probabilities, see previous comment) and fire hazard, and carbon stocks/sinks, from 0 to 1. The risk to carbon stocks and sinks was calculated by multiplying the rescaled hazard values by the rescaled carbon stock and sink values, with higher values indicating higher vulnerability.**

**We acknowledge that using quantile-based classification can create a perception of relative risk, always showing 20% of the area at very high risk. To address this, we now present the risk map**

**using values on a scale from 0 to 1. This approach provides a more accurate and clear representation of risk levels.**

- A critical windspeed of 5 m/s as presented in the manuscript is extremely low, and will likely be exceeded every year. In other words, such a stand shouldn't be able to exist. Although the CWS is only used for a relative judgement, it raises questions on how trustworthy the results are. A validation would be nice, or at least a reflection on the CWS that are found.

**Thank you for your feedback on the critical windspeed (CWS) in our manuscript. We reviewed our analysis and noticed that some forest stands had incomplete tree density measurements in the forest management plan (DBH threshold > 17.5 cm). Seemingly low tree densities produced low CWS and overestimated wind hazard. To make sure our findings were solid, we recalculated the CWS using data only from stands where every tree in the measuring plots was included (DBH threshold >7.5 cm). Through this recalculation, we found that the lowest critical windspeed values were around 11 m/s.**

- I have difficulties with the approach of averaging the critical windspeed in case of multiple cohorts within a stand. Low CWS will have a high likelihood of occurring while a high CWS will have a low likelihood, in a very non-linear way. In fact I think you should take the lowest CWS and not the average, as the lowest CWS will determine your risk.

**We understand your concern about the non-linear nature of CWS probabilities. Upon reviewing our methodology, we agree that using the lowest CWS per forest stand provides a more conservative estimate of hazard potential. We chose the taller trees (in our case, always Norway spruce) to determine the CWS for each forest stand.**

- The authors present carbon stock as well as carbon sink as a potential asset at risk, but don't give any judgement or guideline which of the two are more important. Personally I think the carbon stocks are more important. If you lose them, it takes a long time to rebuild. You may harvest high-risk stands with high stocks and use the wood for products etc. A high carbon sink basically means a good growing forest. I don't think it is a good option to harvest those and put trees back in that grow less. Your risk map will show a reduced risk, but I wouldn't call this a climate-smart option.

**Yes, we agree with the reviewer's comment and add this to the discussion:**

**We present both carbon stock as well as carbon sink as potential assets at risk. However, it is necessary to acknowledge the importance of carbon stocks representing long-term carbon reserves that would require significant time to rebuild in case of their loss. Thus, it should be prioritized to maintain carbon stocks, since their reduction (through e.g., a disturbance) may lead to long-term negative impacts on carbon balance (e.g., Law et al., 2004). Harvesting high-risk forest stands with higher carbon stocks and storing carbon in harvested wood products could mitigate the actual risk while ensuring the carbon storage. A high carbon sink reflects a healthy, vigorously growing forest,**

which is important for continuous carbon sequestration. Harvesting these forests and replacing them with slower-growing species would diminish their growth potential and overall carbon sequestration capacity, thus hindering climate mitigation efforts. In the framework of our study, it would be more effective to focus on high-stock forest stands at risk. Additionally, considering shortening the harvest period for younger forests with high carbon sinks at risk may also be beneficial. This approach balances the protection of long-term carbon reserves with the maintenance of continuous carbon sequestration.

- After the detailed risk models are applied, the authors apply simplified regression models to determine what are the underlying risk factors. Not surprisingly, they show exactly the same factors that are entered into the risk models, and that are already described as being important in risk modelling. I think the value of these regression models is that they could be used in other areas, without the need to run the detailed risk models. Showing such an application (plus perhaps validation) would greatly enhance the study.

We added this to the discussion:

The value of the regression models lies in their potential applicability to other areas without the need to run detailed risk models, which require using and understanding specific software. This could provide a practical tool for broader forest management strategies, enabling effective risk assessment and management in diverse forested regions.

- At the same time, I wonder what this study in practice brings for the forest managers of the study area. It tells them that stands with high stocks, tall trees and a high share of spruce are more vulnerable. Isn't this common knowledge? Do they need such an approach to locate such stands? Or would the approach be more suitable to map risk at larger scale?

While it is a common knowledge that stands with high stocks, tall trees, and a high share of spruce are more vulnerable, our study offers added value by quantifying and prioritizing this risk. We provide precise calculations of real probability and exceedance risk. Forest managers can use these calculations to make informed decisions about where to allocate limited resources for risk mitigation and management. Furthermore, when integrating risk probabilities into the calculation of carbon credits, forest managers can better assess the financial viability and sustainability of certain stands. This adds a layer of economic analysis to forest management practices, supporting more robust and financially sound strategies.

We acknowledge the reviewer's comment on scale issues, and add this to the discussion: While the study focuses on specific stands, this approach could be applied to map risks at larger scales. This allows for a broader application in regional and national forest management planning, offering a comprehensive tool for assessing and managing forest vulnerability.

- On the same line, the authors only give very generic advice on what to do if you have a high-risk stand. Does that really apply to all stands? Line 320 includes the advice to avoid unstable edges (which is not included in the modelling approach at all I think), and selective thinning for better anchorage. However, thinning an overstocked tall spruce stand is extremely risky, as it opens up the canopy to the wind, and a total collapse may follow. In practice you may rather clearfell the whole stand, or leave it until something happens.

**We recognize the reviewer's concerns regarding generality of our recommendations and we focused more on the context regarding our study. Regarding the unstable edges, in ForestGALES simulations we used "windfirm edge" for all forest stands for simplification. However, in the discussion we acknowledge that unstable edges might be problematic and cause damages during windthrow events.**

**We rewrote a part of the discussion:**

**For simplification purposes, we did not account for different forest edges in our simulations, setting this parameter to a constant – "windfirm edge." Nevertheless, we acknowledge that unstable edges may cause damage, and possible treatments include feathering (i.e., edge thinning to decrease wind loads and preserving trees that are more likely to withstand windthrow damage). In our case, improving forest edges would include favouring more resistant tree species like beech. It is also recommended to avoid a new forest edge, as the trees growing within the stand are less adapted to higher levels of exposure. Harvesting should avoid creation of new edges and use already established edges, as e.g., a forest road. Forest stand should be managed to grow more stable trees with larger diameters and better root system, which can be a result of a good selective thinning (Mason and Valinger, 2013). However, selective thinning in spruce stands must be done in earlier stand age, otherwise, for a period of time, it destabilizes the treated forest stand that may completely collapse (Albrecht et al., 2012).**

*Albrecht, A., Hanewinkel, M., Bauhus, J., & Kohnle, U. (2012). How does silviculture affect storm damage in forests of south-western Germany? Results from empirical modeling based on long-term observations. European Journal of Forest Research, 131(1), 229–247. https://doi.org/10.1007/s10342-010-0432-x*

*Mason, B., & Valinger, E. (2013). Managing forests to reduce storm damage. In B. Gardiner, A. Schuck, M.-J. Schelhaas, C. Orazio, K. Blennow, & B. Nicoll (Eds.), Living with Storm Damage to Forests What Science Can Tell Us (Vol. 3, pp. 87–96). European Forest Institute.*

- For among others the above reasons, I think there is absolutely no ground for the statement in line 283/284 that "we showed that hazard modelling can be a valuable support…".

**With our improved calculation of windthrow probabilities we demonstrated that hazard modelling can be a valuable support, especially for prioritisation of preventive silviculture, adding a significant value to the risk analysis. Although the regression modelling analysis alone does not provide real probabilities, it may be sufficient for forest managers.**

- To me it is unclear how much of the characteristics of the forest/the stands is really measured, and how much is averaged/interpolated and how much is taken from the forest management plans. One single random plot (line 97) in a stand is by no means enough to characterise the stand. Maybe a map could be included that shows what part of the area was measured, what was interpolated and what was taken from management plans or otherwise.

**We acknowledge the lack of clarity regarding the data sources and have revised the methods section to make this more explicit.**

**At Fusine, we used data from forest management plans to characterize the stands for ForestGALES simulations. Field data were used to validate the forest management plan data, which were collected using LiDAR. At Galeata we used a combination of data from forest management plans (field-measured) together with our field data (i.e., crown base height) to provide details per species and diameter class. The stands characteristics served as an input data for FlamMap simulations.**

- The protocol as described in line 102-124 gives rise to many questions:
- Line 105 states that all pieces of deadwood of 2.5-7.5 cm are counted. Line 110 applies a correction factor of 1.13 for diameters smaller than 7.5 cm, but these have been counted already? I thought perhaps the authors mean 2.5 cm, but line 121 shows that the fuel load for deadwood smaller than 2.5 cm is measured separately. Why the correction factor?

**The line-intersect sampling estimator assumes that coarse woody debris (CWD) pieces lie horizontally on the ground. If pieces are sloped or partially hanging, the length of each piece should be multiplied by the cosine of the angle between the piece and the horizontal plane. This correction factor ranges from 1 (for horizontal pieces) to a maximum of 1.41 (for a 45-degree lean). The correction factor is always 1 or greater, so ignoring it leads to underestimating the target parameter. For deadwood smaller than 7.5 cm, Brown (1982) sets the correction factor to 1.13 by default.**

*Brown, J. K., Oberheu, R. D., & Johnston, C. M. (1982). Handbook for Inventorying Surface Fuels and Biomass in the Interior West.*

- Line 106 states that the amount of deadwood is calculated in t/ha, while equation 1 results in the volume (V) deadwood in m3/ha.

**To clarify this, we provided a more detailed explanation in the methods section. To calculate the deadwood mass (t/ha), we multiplied the volume by a specific density of 440 kg/m³.**

- Line 115 indicates that the number of shrubs intersecting the transect was used, while in line 104 only the height of shrubs was measured at 1 m intervals. When is a shrub considered to intersect with a transect? Only at the 1 m intervals or always? What happens if the same shrub has a canopy that extends over multiple 1 m intervals? How do you estimate shrub density per ha from the number intersects?

**Thank you for your comments. In response to your queries, we adjusted the methods. We have clarified in the methods section that we also measured the number of shrubs intersecting the**

transect. Shrubs were measured only at 1 m intervals along the transect. If a shrub intersects the transect at more than one 1 m interval, it is counted as multiple shrubs. The density of shrubs per ha is calculated as $10\,000/d^2$, where d is the mean distance between shrubs determined as the total distance of the three intersects (30 m) divided by the number of shrubs intersected.

---

## Referee Report (RR1)

General comment

I think the authors did a good job replying to and implementing the reviewers' comments. I am now satisfied with the manuscript, which in my opinion deserves publication. You can find few technical corrections and minor comments below.

Technical corrections

26-28 I would combine the two sentences, as they are roughly saying the same thing.

72-73 Is there any climate data available for Tarvisio covering a 30-year period?

81-82 Could you add something more specific about size and severity of windthrows, if reported by the management plans?

82-83 Would rewrite as "However, extraordinary cuts (…) amount**ed** to 30% of the **total harvested volume** in the years 1997-2019."

84-85 Could you be more specific about the "significant volume reported as extraordinary cuts".

101 Would modify to "given the **scarce** rainfall, summer **droughts**, …".

254-255 I would put the definition of "slenderness" at line 248, where the word appears the first time.

373 remove "it should".

402 Could you be more specific when saying that "selective thinning in spruce stands must be done in earlier stand age"? Which stand age, or earlier than which age?

431-434 I would rather explicitly refer to Testo Unico Forestale here, than a general "In the last 30 years …". Furthermore, I find the final sentence (lines 433-4) rather superfluous, I would remove it.